# Comparing Homogenized Instantaneous Meals and Traditional Pureed Foods in Patients Affected by Dysphagia: A Pilot Study [note 1]

**DOI:** 10.3390/jcm13113160

**Published:** 2024-05-28

**Authors:** Samir Giuseppe Sukkar, Giulia Lorenzoni, Alice Carraro, Francesca Angioletti, Dario Gregori

**Affiliations:** 1Dietetics and Clinical Nutrition Unit, Ospedale Policlinico “San Martino” IRCCS, 16132 Genova, Italy; samir.sukkar@hsanmartino.it (S.G.S.); dietista.alicecarraro@gmail.com (A.C.); 2Unit of Biostatistics, Epidemiology and Public Health, Department of Cardiac, Thoracic, Vascular Sciences and Public Health, University of Padova, 35131 Padova, Italy; giulia.lorenzoni@unipd.it; 3Zeta Research Srl, 34129 Trieste, Italy; francescaangioletti@zetaresearch.com

**Keywords:** malnutrition, dysphagia, modified texture food

## Abstract

**Background:** Institutionalized individuals with dysphagia are particularly at risk for malnutrition. This study investigated two texture-modified models for patients with dysphagia, as follows: (i) traditional homemade pureed food (PF) and (ii) homogenized meals obtained from dehydrated and rehydrated instantaneous preparations (IPs). **Methods:** A retrospective pilot study was performed. It included patients affected by medium-severity dysphagia admitted to the nursing home “Sacra Famiglia” Institute of Cocquio Trevisago, Varese. The patients were aged 41–81 years old and all had complex disabilities. They underwent anthropometric and biochemical parameter assessments at baseline, as well as at two months and four months follow-up. **Results:** The study involved 30 patients, 15 received the IP meal. The comparison between the baseline and the follow-up did not show significant anthropometric and biochemical parameter differences. Conversely, the IP group reported significantly higher levels of consumption and satisfaction, evaluated using a modified Chernoff scale based on three levels of smiles, than the PF group. **Conclusions:** The present findings provide promising indications to improve the diet of patients affected by dysphagia, since meal satisfaction is a relevant factor that has been shown to be associated with better patient mood, motivation to eat, and adherence to prescribed diet.

## 1. Introduction

Dysphagia is defined as having difficulty swallowing, which can lead to inadequate nutrient intake and an increased risk of malnutrition. Institutionalized people, such as those living in nursing homes or hospitals, are particularly vulnerable to malnutrition [1] due to several factors, including age-related changes, cognitive impairment, and medical conditions that can impact their ability to eat and drink.

The prevalence of dysphagia in institutionalized individuals is high, with estimates involving at least one-third of institutionalized subjects depending on the population under study [2]. Dysphagia can result from a wide range of medical conditions, including stroke, Parkinson’s disease, dementia, and head and neck cancer. In addition, the aging process itself can lead to dysphagia [3]. In turn, a recent systematic review has shown that dysphagia is associated with poor oral health, dehydration, and increased malnutrition risk [4].

Malnutrition, a common result of dysphagia [5], can have severe implications for health outcomes. It can lead to a weakened immune system [6,7], increased susceptibility to infections, and impaired wound healing, among other complications. In addition to that, it can exacerbate underlying medical conditions and increase morbidity and mortality [8].

Institutionalized individuals with dysphagia are particularly at risk for malnutrition, due to several factors [9]. First, they may have limited access to food and liquids or require special diets or nutritional assistance. Second, they may have difficulty chewing or swallowing certain foods or liquids, which may limit their intake of essential nutrients. Third, they may have underlying medical conditions that affect their nutritional status, such as diabetes or kidney disease. Finally, they may have cognitive impairment, which can affect their ability to communicate their needs and preferences, leading to further complications.

Several strategies have been employed to enhance dietary intake among elderly individuals with dysphagia in nursing homes, particularly by enriching the nutrient content of meals. This includes the addition of supplements or enriching agents to food. Further dietary techniques involve recreating the appearance and feel of conventional foods by using thickeners or nutrient enhancers, tailoring the meal’s texture and consistency to the specific requirements of the patients, and providing nutrient-rich snacks between meals. However, no well-established interventions are available in this context, due to a lack of evidence in the field [10]. For example, a recent systematic review and meta-analysis showed that texture-modified food is associated with a lower nutrient intake compared to regular diets [11].

This study investigated two texture-modified food models—(i) traditional homemade pureed food (PF) and (ii) homogenized meals obtained from dehydrated and rehydrated instantaneous preparations (IPs).

## 2. Materials and Methods

### 2.1. Study Population

The present retrospective pilot study included thirty patients affected by medium-severity dysphagia (not candidates for artificial nutrition) who were admitted to the nursing home “Sacra Famiglia” Institute of Cocquio Trevisago, Varese. Patients administered a traditional PF diet were compared to those administered a newly introduced IP diet in the nursing home.

The patients were aged 41–81 years old and all had complex disabilities. Assessment for oropharyngeal motor dysphagia was based on the Smithard oral test. Subjects were excluded if they were in a terminal phase of the illness, had severe dysphagia, were treated with artificial nutrition (enteral/parenteral), or had severe renal, hepatic, or respiratory insufficiency.

As this was an observational retrospective pilot study, the Institutional Review Board was notified of this, according to Italian regulations, and waived formal approval.

### 2.2. Meals’ Characteristics

IP meals were “IoSano”^®^ instantaneously homogenized meals based on standardized recipes. A specialized machinery was used to implement the IP diet. It was installed in the nursing home’s kitchen and used by the trained staff of the nursing home. The configuration of the machinery and service formats were periodically customized based on the specific menu. The nutritional values of the IP menu were as follows: 1688 ± 224 kcal; fat: 84.66 ± 10 g; total saturated fats: 18 ± 4 g; carbohydrates: 173 ± 48 g; sugar: 53 ± 4 g; total fiber: 14 ± 2 g; protein: 60 ±4 g; sodium: 1467 ± 12 mg.

The PF menu was based on homemade pureed food. Briefly, the PF menu features foods like those served to patients without dysphagia but modified to ensure safety, especially through changes in texture. The nutritional data were as follows: 1709 ± 254 kcal; fat: 93 ± 15 g; total saturated fats: 22 ± 3 g; carbohydrates: 157 ± 35 g; sugar: 58 ± 4 g; total fiber: 15 ± 2 g; protein: 61 ± 3 g; sodium: 1704 ± 15 mg.

Recommended nutrient levels in both menus were calculated according to the Italian Recommended Energy and Nutrient Intake Levels for a level of physical activity of 1,4 (levels associated with minimally active, sedentary lifestyle).

### 2.3. Data Collection

Data were collected at baseline and at each follow-up (at two months (FU2) and four months (FU4)).

At baseline, patients underwent the evaluation of physical functioning, using the Activities of Daily Living (ADL) scale, and malnutrition risk, using the Malnutrition Universal Screening Tool (MUST) [12].

Anthropometric parameters, including weight, height, and bioimpedance parameters (phase angle and body composition data), were evaluated at baseline and each follow-up, together with biochemical parameters (azotemia and albumin). All patients also underwent an evaluation of malnutrition risk using the MNA-SF (mini nutritional assessment short form) at baseline and each follow-up [13].

At follow-up, satisfaction with the diet was evaluated using a modified Chernoff scale based on three levels of smiles (higher scores indicated lower levels of satisfaction). Consumption levels were evaluated using a 3-point Likert scale, where higher scores meant higher consumption levels. Signs and symptoms of dysphagia were assessed at each meal occasion and the weekly frequency was reported (0 = never, 1 = once a week, 3 = 1–2 times per week, 10 = 1–2 times per day, 14 = always), with a score of 0 indicating the absence of symptoms. Finally, the safety of the diets was evaluated considering the occurrence of aspiration pneumonia and infections at follow-up.

### 2.4. BIA Analysis

BIA was performed with a single tetrapolar BIA measurement of resistance (R) and reactance (Xc), directly measured in Ohms at 50 kHz, 800 μA, at a fixed frequency of 50 kHz between the right wrist and ankle (standard placement of surface electrodes) with a body impedance analyzer (BIA 2000-S; Akern srl, Firenze, Italy), while the subjects were in a prone position on a nonconductive surface. The determination of body composition parameters (Extracellular Water (L), Phase Angle, Fat Mass (kg), and Fat-free Mass (kg)) was performed according to the literature [14].

### 2.5. Power Analysis

As a pilot retrospective study, the design did not allow for formal power analysis or sample size calculation. Nevertheless, as a general indication, we computed the power that could be obtained for a reasonable set of scenarios. Since most of the response variables were continuous, the power analysis referred to the difference between strategies at FU4 using a *t*-test and an alpha level of 0.05. Again, given the observational and pilot structure of the study, no adjustments for a multiplicity of testing were performed at either the planning or analysis stages. For subjects between 12 and 20 per group, at a power level of 0.80, effect sizes of about ±0.2 were detectable.

### 2.6. Statistical Analysis

The data were described using median values (I, III quartile as measures of variability) or mean ± standard deviation, whichever was appropriate, for continuous variables, and percentages (number of cases) for categorical variables. Tests for the difference between strategies at each time point (baseline, FU2, and FU4) were based on the Wilcoxon test or *t*-test, whichever was appropriate, for continuous variables and the Chi-Square test for categorical variables. The *p*-value for trend was computed using a linear model for each variable as a response, with time as a continuous variable, using the Wald test. Analyses were performed using the R System and the rms libraries.

## 3. Results

### 3.1. Baseline Characteristics

Fifteen patients were administered with the IP diet. The physical functioning was poor in both groups, with median values of 1 and 0.5 in the IP and PF groups, respectively (*p*-value 0.624). No significant differences were detected at baseline in the malnutrition risk, evaluated using the MNA-SF (Figure 1), and in the BMI (IP BMI: 20.37; PF BMI: 25.10; *p*-value 0.064). Regarding body composition, fat-free mass and extracellular water were lower in the IP group (Table 1).

### 3.2. Trends over Time and Differences between Strategies during Follow-Up

The comparison between the baseline (T1) and the follow-up did not show significant differences in the parameters of interest.

Significant differences between groups were detected at each follow-up for weight (higher in the PF group, *p*-value < 0.001), extracellular water (higher in the PF group, *p*-values of 0.009 at FU2 and 0.004 at FU4), and fat-free mass (*p*-values of 0.003 at FU2 and <0.001 at FU4). Conversely, no significant differences between groups were detected in the levels of azotemia at follow-ups.

### 3.3. Consumption, Satisfaction, and Symptoms

Throughout the entire follow-up period, the IP group reported significantly better consumption and satisfaction (for satisfaction, lower scores mean higher satisfaction) than the PF group (PF satisfaction: 1.16 ± 0.41; PF consumption: 0.94 ± 0.18; IP satisfaction: 1.07 ± 0.33; IP consumption: 0.96 ± 0.17; both *p*-values < 0.001). This trend was also observed when the analysis was stratified by meal occasion (Table 2). Furthermore, the IP group consistently reported a significantly lower frequency of signs and symptoms of dysphagia than the PF group during meals (Table 3).

No infections and aspiration pneumonia cases were recorded during the follow-up period.

## 4. Discussion

The nutritional composition of meals for institutionalized dysphagic patients is a crucial component in their management, to ensure adequate dietary intake and promote their overall health. However, it is noteworthy that meal satisfaction and patient preferences are also relevant aspects to be considered [15]. A satisfying meal can improve the patient’s mood, increase motivation to eat and promote adequate nutrient intake.

Satisfaction with meals of institutionalized patients, particularly those with dysphagia, is often an under-explored aspect in clinical research, where the focus is mainly on nutrient intake, quantities consumed, and biochemical and anthropometric parameters. Patient satisfaction is rarely considered as an outcome, despite its potential impact on therapeutic success. In cases of dysphagia, feeding itself is a form of therapy and, understandably, clinical priorities may overshadow soft outcomes such as meal satisfaction, focusing more on critical aspects such as aspiration risk and weight loss.

However, the literature indicates that meal satisfaction can influence adherence to prescribed diets and the patient’s overall quality of life [16,17], which are crucial to the health outcomes of these patients. Unsatisfied patients may seek alternative meals, which may be less healthy or less suitable for their nutritional needs. Improving the sensory qualities of meals can lead to greater satisfaction [18], encouraging better eating behaviors and dietary outcomes. Evaluating and integrating patient preferences and improving meal presentation are recommended as part of diet management in nursing facilities.

### Study Limitations

Even if powered according to realistic scenarios, this pilot study remains characterized by a small sample size. Therefore, findings need to be confirmed preferably using a randomized study design to obtain more robust evidence on the topic.

## 5. Conclusions

Although the results of this pilot study are preliminary, they offer encouraging insights into improving the diets of institutionalized patients with dysphagia. Despite the lack of significant changes in anthropometric and biochemical parameters of patients following two different dietary regimens, an important and often overlooked aspect emerged—satisfaction with meals. This satisfaction is recognized as crucial in encouraging meal consumption and adherence to therapeutical recommendations.

## Figures and Tables

**Figure 1 jcm-13-03160-f001:**
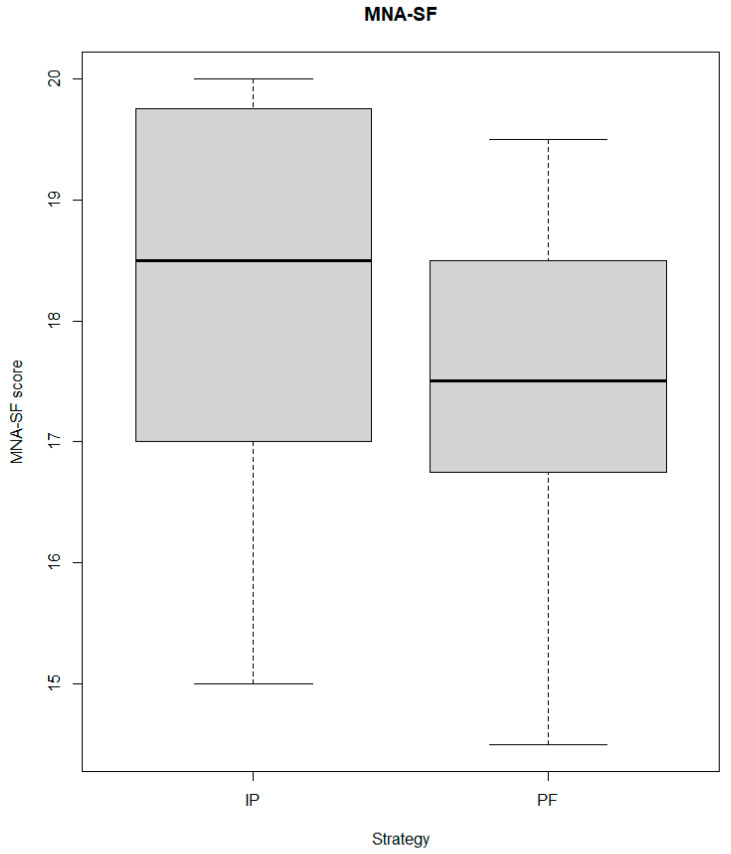
Boxplots presenting the distribution of the MNA-SF score in the two groups.

**Table 1 jcm-13-03160-t001:** Serum markers. Data are I quartile/Median/III quartile for continuous variables and percentage (number of cases) for categorical variables. The *p*-value for the trend indicates the statistical significance of the change over time for each variable. *p*-value between strategies (to be read column-wise) indicates the statistical significance of the median values between the two strategies (IP vs. PF) at each time point (which are compared row-wise). Overall values refer to the overall sample.

		Baseline	FU2	FU4	*p*-Value (for Trend)
Anthropometric and BIA assessment					
BMI	Overall	19.72/21.36/25.85	19.80/20.95/26.53	19.95/20.62/25.55	0.955
	IP	19.23/20.37/21.66	19.00/20.60/21.90	19.01/20.25/21.42	0.981
	PF	20.24/25.10/29.54	20.50/25.30/29.90	20.35/25.10/30.85	0.974
*p*-value (between strategies)		0.064	0.049	0.022	
Weight (kg)	Overall	46.08/55.40/63.63	46.63/55.25/63.75	45.75/54.75/62.93	0.95
	IP	43.50/46.30/54.50	42.50/47.00/55.00	42.25/45.90/54.40	0.986
	PF	56.10/64.50/68.63	56.00/63.00/69.50	56.25/63.50/69.43	0.966
*p*-value (between strategies)		<0.001	<0.001	<0.001	
Hydration	Overall	72.70/73.30/73.80	73.50/75.10/77.70	72.95/73.65/73.88	0.708
	IP	69.28/73.00/73.90	73.35/75.10/77.50	73.35/73.70/75.55	0.829
	PF	73.15/73.40/73.70	73.63/75.70/77.48	71.00/73.50/73.85	0.28
*p*-value (between strategies)		0.536	0.702	0.278	
Extracellular Water (L)	Overall	17.20/18.90/21.20	18.80/20.40/21.70	18.15/19.45/21.58	0.893
	IP	16.10/18.45/18.88	17.55/18.90/20.45	17.30/19.00/19.45	0.997
	PF	18.40/20.80/22.05	20.30/21.85/24.58	19.20/21.40/23.83	0.794
*p*-value (between strategies)		0.006	0.009	0.004	
Phase Angle	Overall	3.50/4.10/4.70	3.50/4.00/4.70	3.63/4.35/5.08	0.891
	IP	3.30/3.50/4.38	3.25/3.50/4.35	3.30/4.10/4.55	0.8
	PF	4.10/4.50/4.90	3.78/4.30/4.63	4.15/4.60/5.55	0.888
*p*-value (between strategies)		0.021	0.112	0.057	
Fat Mass (kg)	Overall	6.40/8.20/13.20	3.60/7.70/13.00	5.03/7.25/10.98	0.994
	IP	5.18/7.25/8.50	3.25/6.70/8.25	4.40/7.20/8.10	0.9
	PF	6.50/11.00/18.85	4.83/10.85/17.75	5.15/7.40/15.55	0.904
*p*-value (between strategies)		0.141	0.107	0.338	
Fat-free Mass (kg)	Overall	23.00/25.90/28.50	23.90/26.90/29.00	23.93/26.30/29.55	0.7
	IP	22.83/24.75/25.85	23.45/25.60/26.95	22.85/24.00/26.30	0.927
	PF	25.05/28.50/29.95	27.20/29.10/31.73	27.00/29.60/30.55	0.943
*p*-value (between strategies)		0.007	0.003	<0.001	
Biochemical parameters and Nutritional assessment					
MNA-SF	Overall	17.00/17.50/19.38	16.63/17.75/19.50	17.50/18.25/19.88	0.309
	IP	17.00/18.50/19.75	17.50/19.50/20.00	18.00/20.00/20.50	0.237
	PF	16.75/17.50/18.50	15.50/17.00/18.00	16.50/17.50/18.50	0.323
*p*-value (between strategies)		0.324	0.004	0.006	
Albumin (mg/dL)	Overall	3.63/3.92/4.13	3.60/3.90/4.25	3.40/3.70/4.20	0.629
	IP	3.34/3.80/3.97	3.40/3.60/4.20	3.50/3.70/4.20	0.849
	PF	3.90/4.00/4.20	3.80/3.95/4.28	3.43/3.70/4.05	0.102
*p*-value (between strategies)		0.067	0.147	0.716	
Azotemia (mg/dL)	Overall	18.75/24.50/31.00	19.75/24.00/34.00	22.00/27.00/35.00	0.061
	IP	17.25/20.50/25.25	19.00/23.00/33.00	21.00/28.00/35.00	0.297
	PF	24.25/28.00/34.75	21.00/29.00/34.00	23.50/27.00/33.50	0.368
*p*-value (between strategies)		0.021	0.669	0.898	

**Table 2 jcm-13-03160-t002:** Satisfaction and consumption levels overall over the entire follow-up period. Data are means and standard deviations. *p*-values refer to the difference between IP and PF groups. Note that for satisfaction, higher scores mean lower satisfaction levels.

	PF	IP	Combined	*p*-Value
Overall				
Satisfaction	1.16 ± 0.41	1.07 ± 0.33	1.12 ± 0.37	<0.001
Consumption	0.94 ± 0.18	0.96 ± 0.17	0.95 ± 0.17	<0.001
Acquagel				
Dissatisfaction	1.13 ± 0.38	1.18 ± 0.52	1.16 ± 0.45	0.79
Consumption	0.95 ± 0.15	0.92 ± 0.23	0.94 ± 0.20	0.28
Dinner				
Dissatisfaction	1.23 ± 0.48	1.06 ± 0.26	1.14 ± 0.39	<0.001
Consumption	0.92 ± 0.20	0.97 ± 0.15	0.95 ± 0.18	<0.001
Breakfast				
Dissatisfaction	1.09 ± 0.31	1.03 ± 0.19	1.06 ± 0.26	0.004
Consumption	0.96 ± 0.13	0.98 ± 0.11	0.97 ± 0.12	0.037
Snack				
Dissatisfaction	1.18 ± 0.44	1.05 ± 0.29	1.11 ± 0.37	<0.001
Consumption	0.95 ± 0.15	0.96 ± 0.17	0.96 ± 0.16	0.091
Lunch				
Dissatisfaction	1.17 ± 0.40	1.05 ± 0.27	1.11 ± 0.35	<0.001
Consumption	0.91 ± 0.21	0.96 ± 0.17	0.94 ± 0.19	0.004

**Table 3 jcm-13-03160-t003:** Symptoms score according to the treatment (IP vs. PF). Data are mean ± SD. *p*-values refer to the difference between groups.

	PF	IP	Combined	*p*-Value
Avoid meals	0.27 ± 0.52	0.27 ± 1.13	0.27 ± 0.90	0.083
Nasal reflux	0.04 ± 0.19	0.04 ± 0.32	0.04 ± 0.27	0.4
Cough	1.10 ± 1.23	0.74 ± 2.19	0.91 ± 1.82	<0.001
Increased secretion	0.91 ± 1.31	0.53 ± 2.15	0.71 ± 1.82	<0.001
Hoarseness	0.45 ± 0.76	0.17 ± 0.82	0.30 ± 0.80	<0.001
Gargling voice	0.12 ± 0.43	0.11 ± 1.04	0.12 ± 0.81	0.014
Cough after meals	1.15 ± 1.12	0.77 ± 2.27	0.95 ± 1.84	<0.001
Clearing throat	0.38 ± 1.18	0.16 ± 0.81	0.26 ± 1.01	<0.001
Alteration of facial expression	3.95 ± 3.93	0.85 ± 2.07	2.29 ± 3.44	<0.001
Delay to start swallowing	0.46 ± 1.93	0.64 ± 2.48	0.56 ± 2.24	0.57
Oral and nasal regurgitation	0.10 ± 0.35	0.167 ± 1.26	0.13 ± 0.95	0.11
Packing food in cheeks	0.77 ± 2.55	0.28 ± 1.60	0.51 ± 2.11	0.002
Multiple swallowing for one bite	0.68 ± 2.21	0.23 ± 1.45	0.44 ± 1.86	<0.001

## Data Availability

The data are available on reasonable request to the corresponding author.

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
