# Peer review of "Comparing Homogenized Instantaneous Meals and Traditional Pureed Foods in Patients Affected by Dysphagia: A Pilot Study†"

_jcm, 2024, doi:10.3390/jcm13113160_

Round 1

Reviewer 1 Report

Comments and Suggestions for Authors

The title does not indicate that this is a medical study. It gives an impression to the reader that this study is about the nutritional comparison of those two types of meals.

Line 20 the phrase underwent OI is awkward. Instead use “the participants received the meal”.

Line 23 there is the mention of satisfaction - how was this measured

line 25 since meal satisfaction is a relevant factor for the well-being of institutionalised subjects is not clear

introduction

this section does not include the references that are essential in this area. Are there any similar studies? What is the justification for research? How widely PF and all I used in institutions?

The coding for the rehydrated instantaneous preparations could be IP rather than or OI, because the latter is slightly confusing.

Materials and methods

line 69 the participants to give consent - this was later stated as it was waived (Line 211). Which one is the correct statement

line 76 what is meant by the kitchen equipment? Do you refer to a company that sells kitchen equipment

line 78 the sentence the distribution formats and equipment were tailored to specific menus and needs is unclear.

Line 83 the the term bromotological is very uncommon. I had to check this word and it means relevant to food science or nutrition. Please use simply and the phrase nutritional value or nutritional profile

line 82 the PF menu - there is no indication what either of those diets or meals entail

line 87 - the acronyms used here need to be given in full because these seem to be the terminology from Italian language

line 87 physical activity level is 1.4. Does this align with the activity level of the participants

Line 103 - BIA should be given in full which is bioelectrical impedance analysis

line 110 Kg should be written as kg

line 132 this line should be removed as this doesn't belong to the result section

line 136 there is a discrepancy in the way that the decimal places are presented there are some with four decimal places and somewhere 3 decimal places

line 137 the word pathognomonic is uncommonly used. Please use commonly used alternatives

table one under the dysphagia section, the acronyms OI and PF should be added as these are missinh

table one not sure what the acronym MNA stands for

table one pneumonia and pressure ulcers have not been mentioned before in the text

table one nutritional risk - how was this measured

table one this table spreads over 3 pages it needs to be better presented so now it will look more professional

line 161 how was the level of consumption measured

discussion

lines 172 to 174 - this is the repetition of the results already mentioned

Line 177 the phrase “we should not forget” is not appropriate for scientific writing

line 185 how was the safety of the oh I meals were determined

conclusion

this section needs to be rewritten and enhanced. Currently there is no mention of the data obtained and the conclusions drawn. The statements are vague and unclear.

References

the reference list can be expanded to include relevant literature. If there isn't any relevant literature, then this should be mentioned in the introduction section. This will corroborate  the justification for this research.

The whole manuscript needs improvement, scientific content wise and the way English is used.  There should be robust methodology included. The statements should be more clear and supported by scientific findings.

Comments on the Quality of English Language

Needs major improvement

Author Response

  • The title does not indicate that this is a medical study. It gives an impression to the reader that this study is about the nutritional comparison of those two types of meals.

The title has been revised according to the reviewer’s suggestion.

  • Line 20 the phrase underwent OI is awkward. Instead use “the participants received the meal”.

Done. The sentence has been revised according to the reviewer's suggestion.

  • Line 23 there is the mention of satisfaction - how was this measured

It was measured using a modified Chernoff scale as specified in the Methods section. The concept has been clarified also in the abstract.

  • line 25 since meal satisfaction is a relevant factor for the well-being of institutionalised subjects is not clear

Done. The concept has been clarified.

introduction

  • this section does not include the references that are essential in this area. Are there any similar studies? What is the justification for research? How widely PF and all I used in institutions?

The introduction has been revised addressing the questions of the reviewer.

  • The coding for the rehydrated instantaneous preparations could be IP rather than or OI, because the latter is slightly confusing.

Done. The original acronym has been replaced with the one proposed by the reviewer.

Materials and methods

  • line 69 the participants to give consent - this was later stated as it was waived (Line 211). Which one is the correct statement

We thank the reviewer for pointing out the inconsistency, which we have resolved as there was an oversight in the list of exclusion criteria.

  • line 76 what is meant by the kitchen equipment? Do you refer to a company that sells kitchen equipment

The methods employed for IP preparation have been clarified in the method section of the manuscript.

  • line 78 the sentence the distribution formats and equipment were tailored to specific menus and needs is unclear.

The preparation method has been clarified in the manuscript.

  • Line 83 the the term bromotological is very uncommon. I had to check this word and it means relevant to food science or nutrition. Please use simply and the phrase nutritional value or nutritional profile

Done. The word has been replaced as suggested by the reviewer.

  • line 82 the PF menu - there is no indication what either of those diets or meals entail

The PF menu features foods similar to those served to patients without dysphagia, but modified to ensure safety, especially through changes in texture. In general, since this is a nursing home, extreme care is taken with all meals provided to residents. Even for those without dysphagia, general principles for prevention of choking are applied, including preference for softer food textures, avoiding meats and cold cuts with filaments and fat, and boning chicken. The concept has been reported also in the methods section.

  • line 87 - the acronyms used here need to be given in full because these seem to be the terminology from Italian language

Done. The acronyms have been clarified.

  • line 87 physical activity level is 1.4. Does this align with the activity level of the participants

Yes, LAF levels of 1.4 are associated with minimally active lifestyles, reflecting the physical activity levels (sedentary) of our patient population. The concept has been clarified in the Methods section.

  • Line 103 - BIA should be given in full which is bioelectrical impedance analysis

Done

  • line 110 Kg should be written as kg

Done

  • line 132 this line should be removed as this doesn't belong to the result section

Done

  • line 136 there is a discrepancy in the way that the decimal places are presented there are some with four decimal places and somewhere 3 decimal places

Done. Decimal places were revised.

  • line 137 the word pathognomonic is uncommonly used. Please use commonly used alternatives

Done.

  • table one under the dysphagia section, the acronyms OI and PF should be added as these are missinh

Done

  • table one not sure what the acronym MNA stands for

It stands for MNA-SF (mini nutritional assessment short form). We would like to help the reviewer for the comment allowing us to clarify the concept on method section.

  • table one pneumonia and pressure ulcers have not been mentioned before in the text

Done. The methods section and the table have been revised to make them clearer.

  • table one nutritional risk - how was this measured

Done. The methods section and the table have been revised to make them clearer.

  • table one this table spreads over 3 pages it needs to be better presented so now it will look more professional

The table has been revised to make it clearer and simpler.

  • line 161 how was the level of consumption measured

The measurement method was specified in the Methods section.

discussion

  • lines 172 to 174 - this is the repetition of the results already mentioned

Line 177 the phrase “we should not forget” is not appropriate for scientific writing

  • line 185 how was the safety of the oh I meals were determined

Aspiration pneumonia was considered an indicator of safety of the meal. The concept has been clarified in the manuscript.

conclusion

  • this section needs to be rewritten and enhanced. Currently there is no mention of the data obtained and the conclusions drawn. The statements are vague and unclear.

Done. The section has been rewritten according to the reviewer’s suggestion.

References

  • the reference list can be expanded to include relevant literature. If there isn't any relevant literature, then this should be mentioned in the introduction section. This will corroborate  the justification for this research.

Done, the introduction section has been revised together with the reference list.

  • The whole manuscript needs improvement, scientific content wise and the way English is used.  There should be robust methodology included. The statements should be more clear and supported by scientific findings.

The full manuscript underwent contents and language revision.

Reviewer 2 Report

Comments and Suggestions for Authors

The manuscript is dealing with patients with dysphagia and their malnutrition problem. The introduction speaks about these, but there is no information from the available traditional and the instantaneous meals at the moment. There is no information about their nutrition value also.

The results are given just in text and tables. May be it can be improved with some figures.

In the tables +/- should be replaced with ± character.

The conclusion is very poor, just show an indicated general result.

Comments on the Quality of English Language

The quality of the English language is enough good.

Author Response

  • The manuscript is dealing with patients with dysphagia and their malnutrition problem. The introduction speaks about these, but there is no information from the available traditional and the instantaneous meals at the moment. There is no information about their nutrition value also.

The nutritional facts of the two dietary regimens have been reported in the Methods section of the manuscript.

  • The results are given just in text and tables. May be it can be improved with some figures.

Done.

  • In the tables +/- should be replaced with ± character.

Done.

  • The conclusion is very poor, just show an indicated general result.

Done. The conclusion section has been revised.

Round 2

Reviewer 2 Report

Comments and Suggestions for Authors

All of my issue were corrected.